# Differences in the Evaluation of Malnutrition and Body Composition Using Bioelectrical Impedance Analysis, Nutritional Ultrasound, and Dual-Energy X-ray Absorptiometry in Patients with Heart Failure

**DOI:** 10.3390/nu16101535

**Published:** 2024-05-20

**Authors:** Ana Benitez-Velasco, Carlos Alzas-Teomiro, Carmen Zurera Gómez, Concepción Muñoz Jiménez, José López Aguilera, Manuel Crespin, Juan Antonio Vallejo-Casas, María Ángeles Gálvez-Moreno, María José Molina Puerta, Aura D. Herrera-Martínez

**Affiliations:** 1Maimonides Institute for Biomedical Research of Cordoba (IMIBIC), 14004 Cordoba, Spain; 2Nuclear Medicine Service, Reina Sofia University Hospital, 14004 Cordoba, Spain; 3Endocrinology and Nutrition Service, Reina Sofia University Hospital, 14004 Cordoba, Spain; 4Cardiology Service, Reina Sofia University Hospital, 14004 Cordoba, Spain

**Keywords:** heart failure, malnutrition, body composition, prognosis

## Abstract

Background: Although malnutrition is frequently observed in patients with heart failure (HF), this diagnosis should be performed carefully since HF itself is associated with increased inflammatory activity, which affects body weight, functionality, and some nutritional parameters; thus, its isolated interpretation can erroneously identify surrogate markers of severity as markers of malnutrition. In this context, we aimed to evaluate the prevalence of malnutrition using different classification systems and perform a comprehensive nutritional evaluation to determine the reliability of different diagnostic techniques. Patients and methods: Eighty-three patients with a recent hospital admission due to HF were evaluated. GLIM diagnosis criteria and subjective global assessment (SGA) were performed; a comprehensive anthropometric, functional, and biochemical nutritional evaluation was performed, in which bioelectrical impedance analysis (BIA), nutritional ultrasound, and dual-energy X-ray absorptiometry (DXA) were performed. Additionally, mortality and additional admissions due to HF were determined after a mean follow up of 18 months. Results: Malnutrition according to the GLIM criteria (54%) accurately distinguished patients with impaired functionality, lower lean mass, skeletal mass index, and appendicular muscle mass (BIA), as well as lower trunk fat mass, trunk lean mass, fat-free mass (DXA), and decreased albumin and increased C-reactive protein serum levels. According to SGA, there were significant changes in body composition parameters determined by BIA, muscle ultrasound, and functional tests between well-nourished patients and patients with risk of malnutrition (53.7%) or who had malnutrition (7.1%), but not when the last two groups were compared. BIA and DXA showed strong correlations when evaluating muscle and fat mass in HF patients, but correlations with nutritional ultrasound were limited, as well as functional tests. A multivariate analysis showed that no significant association was observed between body composition and mortality, but preperitoneal fat was associated with an increased risk of new hospital admissions (OR: 0.73). Conclusions: GLIM criteria identified a lower percentage of patients with HF and malnutrition compared with SGA; thus, SGA could have a role in preventing malnutrition in HF patients. Nutritional evaluation with BIA and DXA in patients with HF showed reliable results of body composition parameters in HF, and both help with the diagnosis of malnutrition according to the GLIM or SGA criteria and could provide complementary information in some specific cases.

## 1. Introduction

Heart failure (HF) currently affects more than 64 million people worldwide. Currently, is considered a global public health issue due to its social and economic impact. Its incidence seems to be stabilized but its prevalence has increased due to population aging, effective evidence-based therapies for HF, and improved survival after ischemic heart disease [1].

Malnutrition is characterized by the presence of deficiencies or excesses in nutrient intake, an imbalance of essential nutrients, or impaired nutrient utilization; it is associated with worse clinical outcomes in different clinical scenarios, including HF. Specifically, malnutrition has been reported in 18.6% of patients with HF and includes patients with preserved left ventricular ejection fraction (LVEF) and reduced LVEF (23% and 15.9%, respectively) [2]. Furthermore, some studies have suggested that decreased body mass index (BMI), hypoalbuminemia, hypocholesterolemia, or lymphopenia are independent predictors of mortality in patients with chronic HF during long-term follow up [3].

In this context, it seems essential to screen malnutrition in all patients with HF. However, any isolated nutritional parameter allows an accurate assessment of nutritional status. Moreover, these parameters may be affected by the HF itself; specifically, these patients present with congestion, neurohormonal activation, and inflammatory activity, which could be responsible for weight loss. Additionally, lymphopenia, hypoalbuminemia, and hypocholesterolemia could be also present due to different and unknown mechanisms [4]. For all these reasons, the isolated interpretation of some parameters can erroneously identify surrogate markers of severity as markers of malnutrition; consequently, it is essential to determine the best clinical method that accurately evaluates malnutrition in patients with HF. Based on this, we aimed to evaluate the prevalence of malnutrition in patients with HF using the current GLIM criteria [5] and the subjective global assessment (SGA). Additionally, we performed a comprehensive morphofunctional nutritional evaluation, which included several techniques for evaluating body composition parameters. This evaluation allowed a complete comparison of the patients according to both classification systems but also permitted an extensive comparison among them in order to determine their accuracy and reliability in patients with HF. Furthermore, additional admissions due to HF and mortality were assessed during a follow up of a mean of 18 months.

## 2. Material and Methods

### 2.1. Patients

This study was approved by the Ethics Committee of the Reina Sofia University Hospital (Cordoba, Spain; reference number 5164). It was conducted in accordance with the Declaration of Helsinki and the national and international guidelines. This is a transversal study, wherein written informed consent was signed by every individual before inclusion in this study. Patients of both sexes aged >18 to <85 with a hospital admission due to HF in the previous 30 days were included. Exclusion criteria were end-stage kidney and/or liver disease. Eighty-three patients were included. Consecutive patients for 12 months were evaluated and invited to participate, and only patients who accepted and signed the informed consent were included.

### 2.2. Study Design

Physical examination included anthropometric information (abdominal, arm, and calf circumferences), body composition analysis [bioelectrical impedance analysis (BIA)], functional tests (up-and-go test and handgrip strength), and nutritional ultrasound of the abdominal adipose tissue and the rectus femoris (RF) muscle of the quadriceps. Additionally, a dual-energy X-ray absorptiometry (DXA) was performed in thirteen patients. Specifically, BIA was performed using a 50 kHz phase-sensitive impedance analyzer (BIA 101 Whole Body Bioimpedance Vector Analyzer, AKERN, Florence, Italy) that delivers 800 µA using tetrapolar electrodes positioned on the right hand and foot. BIA measurements were obtained with the patient in a supine position after five minutes of rest. BIA emphasizes the position of the impedance vector, derived from resistance and reactance values generated from a sex-specific healthy reference population [6]. BIA measurements of patients were standardized for sex and age using data from healthy Italian adults [7]. Measurements include phase angle [(PA) expressed in degrees as arctan (Xc/R) × (180°/*π*) [7]; individual standardized PA value (SPA, which is sex- and age-adjusted when matched to the reference population value [8]); hydration parameters; specifically, fluid percentage within the fat-free mass (FFM expressed in kg); intracellular water (ICW, expressed in kg) and total body water (TBW, expressed in kg)]; nutrition status [body cell mass (BCM, expressed in kg); FFM index (FFMI, expressed in %); fat mass (FM, expressed in kg); FM index (FMI, expressed in %); lean mass (LM, expressed in kg); skeletal muscle mass (SMM, expressed in kg); appendicular skeletal muscle mass (ASMM, expressed in kg); and SM index (SMI, expressed in %)].

As previously mentioned, functional tests included the evaluation of handgrip strength and the timed up-and-go test (TUG). For handgrip strength, a Jamar® hydraulic dynamometer was used (Asimow Engineering Co., Los Angeles, CA, USA). It was measured in a seated position with the elbow flexed at 90 degrees in the dominant hand. The median value of three maximal isometric contractions was used. The TUG consists of a seated patient who gets up, walks 3 m, turns around, walks another 3 m, and sits back down; the used time is measured in seconds.

Ultrasound was performed using a 10 MHz probe and a multifrequency linear matrix (Mindray Z60, Madrid, Spain). RF ultrasound evaluation was performed in a supine position, and evaluated variables included RF anteroposterior and transversal *x*-axis, RF circumference and cross-sectional area (RF-CSA), and subcutaneous adipose tissue of the leg. These measurements were determined without compression in the lower third from the superior pole of the patella and the anterior superior iliac spine [9]. Abdominal adipose tissue ultrasound was also performed in a supine position at the midpoint between the xiphoid appendix and the navel. Specifically, abdominal subcutaneous adipose tissue (AT) and preperitoneal AT were determined; data were measured in centimeters [10].

Finally, a DXA scan was performed using a Prodigy Full Size, General Electric Bone and a Mineral Health densitometer with 4.5° narrow fan X-rays (WI, USA), software enCORE^®^ v. 18 (WI, USA). The arms, legs, and trunk are separately evaluated, as well as the right or left side of the body; specifically, total, fat, lean, and fat-free mass were determined, and bone mineral content (CMO), the relative skeletal muscle mass index (RSMMI), appendicular lean mass (ALM), and gynoid or android distribution of the body compartments were also reported.

The presence of malnutrition was evaluated according to the GLIM criteria [5] and, additionally, the subjective global assessment (SGA) was used. The SGA is a nutrition assessment tool that uses structured clinical parameters to diagnose malnutrition. It classifies patients into three categories: well nourished, at risk of malnutrition, and malnutrition [11].

### 2.3. Statistical Analysis

Continuous variables were expressed as the median with an interquartile range (IQR), and categorical variables were described as proportions. For specific group analysis, the absolute number has also been expressed in brackets. Between-group comparisons were analyzed by the Mann–Whitney U test (nonparametric data). A chi-squared test was used to compare categorical data. Using multivariate analysis, the odds ratio (OR) with an interquartile range was calculated. Statistical analyses were performed using SPSS statistical software version 20 and Graph Pad Prism version 6. *p*-values < 0.05 were considered statistically significant. 

## 3. Results

### 3.1. Baseline Characteristics of the Groups

Eighty-three patients were included. Most of them were male (75.9%), with a median age of 65.7 years (Table 1). According to the GLIM criteria, 54.2% of patients presented with malnutrition; in contrast, using the SGA classification, 39.7% were considered well nourished, 53% were at risk of malnutrition, and 7.3% had malnutrition. We did not observe significant differences in the number of additional HF admissions or mortality (Table 1).

According to the GLIM-based classification, patients with malnutrition presented with a higher prevalence of weight loss, lower food texture modifications, and lower capacity for physical activity (*p* < 0.05; Table 1). When the SGA classification was used, weight loss was more prevalent in patients at risk of malnutrition or patients who were malnourished; specifically, patients at risk lost about 3 Kg in the previous 6 months compared with 6.5 kg in the malnutrition group (*p* < 0.05; Table 2). Similarly, changes in food texture were more prevalent in these two groups (*p* < 0.05). There were no significant differences in the presence of gastrointestinal symptoms when the three groups were compared; the capacity to perform moderate physical activity was less prevalent in patients with malnutrition or at risk of malnutrition (*p* < 0.05; Table 2). We did not observe significant differences in the number of additional HF admissions or mortality when the three groups were compared (Table 2). Despite differences between both methods, there was a high concordance among them; specifically, all patients classified as without malnutrition according to the GLIM criteria were well nourished, and five of them were at risk of malnutrition according to the SGA classification; all patients with malnutrition were identified with both classification methods. 

### 3.2. Morphofunctional Evaluation Using Bioimpedance Analysis, Nutritional Ultrasound, and DXA in Patients with HF and Malnutrition According to the GLIM Criteria

Patients with malnutrition presented with lower body weight (72 kg) and a higher percentage of body weight loss (4%) than patients without malnutrition (82.5 kg and 0.6%, respectively, *p* < 0.05). Despite this, BMI was not statistically significantly lower in patients with malnutrition (26.4 vs. 28.7 kg/m^2^; *p* = 0.09; Table 3). According to the BIA analysis, lean mass, water, ASMM, and SMI were significantly lower in patients with malnutrition (52.3 kg, 38.1 kg, 19.9 kg, and 8.3 cm^2^/m^2^, respectively) compared with patients without malnutrition (71.2 kg, 42.8 kg, 22.2 kg, and 10.2 cm^2^/m^2^, respectively, *p* < 0.05). 

Remarkably, arm circumference was significantly lower in patients with malnutrition (27 cm vs. 30 cm, *p* = 0.02) but not calf circumference (36 vs. 37 cm *p* = 0.21). Using the RF nutritional ultrasound, the RF-CSA and both axes were significantly lower in patients with malnutrition (3.4 cm^2^ vs. 4.1 cm^2^, 1 cm vs. 1.3 cm; 3.7 vs. 4.2 cm^2^); additionally, the subcutaneous adipose tissue of the abdomen tended to be lower in these patients (3 cm vs. 9.6 cm, *p* = 0.06), but there were no statistical differences in the distribution of preperitoneal AT (1.6 cm vs. 3.3 cm, *p* = 0.11). Functional tests revealed that handgrip strength was lower in patients with malnutrition (23 kg) compared with patients without malnutrition (34.8 kg; *p* = 0.002); additionally, the TUG was longer in patients with malnutrition (19 s vs. 15 s, respectively, *p* < 0.001; Table 3).

In patients in which DXA was performed, patients with malnutrition presented significantly lower total and trunk body mass, fat, lean, and fat-free mass (*p* < 0.05). Additionally, the RSMMI tended to be lower in patients with malnutrition (*p* = 0.06), but there were no significant differences when ALM and ALM/height^2^ were analyzed (Table 4).

Regarding biochemical parameters, serum albumin tended to be lower in patients with malnutrition (4.2 g/dL vs. 4.4 g/dL, *p* = 0.07); C-RP was higher in these patients (6.8 vs. 3.7 mg/L; *p* = 0.05), and 25OHvitD was also lower in this group (13 ng/dL vs. 23 ng/dL, *p* = 0.02). Detailed biochemical values are depicted in Table 5.

### 3.3. Morphofunctional Evaluation Using Bioimpedance Analysis, Nutritional Ultrasound, and DXA in Patients with HF and Risk of Malnutrition According to the SGA Criteria

According to the BIA, patients at risk of malnutrition presented lower body weight, BMI, BCMe, ECMe, FM (Kg), LM (Kg), water (Kg), and bone mass (Kg; *p* < 0.05); ASMM tended to be lower in these patients (*p* = 0.06). Arm circumference, calf circumference, and handgrip strength were also lower in these patients, while the TUG increased (*p* < 0.01). According to the nutritional ultrasound parameters, the RF anteroposterior axis was lower in these patients (*p* = 0.02), and the RF transversal axis also tended to be decreased (*p* = 0.09; Table 3). There were no statistically significant differences in the adipose tissue parameters determined by ultrasound or abdominal circumference. Almost all of the same parameters were significantly altered when well-nourished and malnourished patients were compared; additionally, ASMM and the RF transversal axis also decreased (Table 3). There were no statistically significant changes in body composition parameters determined by anthropometry, ultrasound, or BIA, but calf circumference statistically changed when patients at risk of malnutrition were compared with patients with malnutrition (*p* = 0.004) (Table 3).

Finally, in patients in which DXA was performed, significant changes in body composition parameters were not significantly different when the three groups were compared (Table 4).

Regarding biochemical analyses, there were no statistically significant differences in nutritional-related biochemical parameters when the three groups were compared. Only NT-proBNP was significantly increased when well-nourished patients were compared with patients at risk of malnutrition and with patients with malnutrition (*p* = 0.04 in both cases; Table 5). 

Age- and sex-adjusted baseline LVEF, percentage of body weight loss, and arm circumference tended to correlate with mortality. Only control NT-proBNP was associated with increased mortality (OR: 1; *p* = 0.02). Regarding new hospital admissions due to HF, only preperitoneal AT was significantly associated with this parameter (OR: 0.73; *p* = 0.01; Table 6).

### 3.4. Clinical Correlations between Different Body Composition Measurements Determined by BIA and DEXA

BCM positively correlated with arm and leg total mass, arm and leg lean mass, leg CMO, trunk total mass, trunk total fat, trunk lean mass, and CMO. The correlation was stronger with the right side LM (rho = 0.956) than the left side LM (rho = 0.821; Figure 1).

In contrast, SMI negatively correlated with arm FM and positively with arm LM, leg total mass, leg LM, body trunk LM, and body trunk CMO; additionally, the correlation with right or left side LM was similar (rho = 0.696 vs. 0.612, respectively; Figure 1).

FM determined by BIA strongly correlated with FM determined by DEXA, especially with the legs (rho = 0.832) and trunk (rho = 0.859) at both sides, right (rho = 0.888) and left (rho = 0.938). FFM determined by BIA also strongly correlated with total and LM determined by DXA, especially with leg total mass (rho = 0.8), leg lean mass (rho = 0.888), trunk total mass (rho = 0.808), trunk LM (rho = 0.853), trunk CMO (rho = 0.870), total mass from the right and left side (rho = 0.836 and 0.959, respectively), and CMO from the right and left side (rho = 0.818 and 0.815, respectively).

PA positively correlated with arm LM, leg total mass, leg LM, and right side lean mass. SPA did not significantly correlate with any body composition parameter determined by DXA (Figure 1).

Android distribution of the total mass strongly correlated with FM (rho = 0.840) and also with other variables, including TBW, ECW, ICW, and FFM (rho > 0.7); android distribution of the lean mass correlated with ECW (rho = 0.818). In contrast, gynoid distribution of the total mass strongly correlated with TBW, ICW, FFM, FM, BCM, and ASMM (rho > 0.8); gynoid distribution of the LM strongly correlated with TBW, ICW, FFM, BCM, and ASMM (rho > 0.9). As expected, total FM and FFM determined by both methods reflected strong positive correlations (rho = 0.909 and 0.941, respectively), and LM determined with DEXA strongly correlated with FFM determined by BIA (rho = 0.926). Additionally, the RSMMI strongly correlated with BCM (rho = 0.924), as well as with TBW, FFM, and ASMM (rho = 0.876, 0.876, and 0.840, respectively; Figure 2).

### 3.5. Clinical Correlations between Different Body Composition Measurements Determined by Nutritional Ultrasound and DXA

Preperitoneal AT strongly correlated with trunk total mass (rho = 0.739), and other significant correlations were observed between abdominal, superficial, and preperitoneal AT determined by ultrasound and FM parameters determined by DXA, but these correlations were weaker (rho < 0.8; Figure 3A).

When the RF ultrasound was evaluated, RF-AT correlated with right side FM (rho = 0.738) and leg FM (rho = 0.718). The RF-CSA correlated with arm LM (rho 0.714; Figure 3A).

The gynoid distribution of total and FM strongly correlated with preperitoneal AT (0.788 and 0.868, respectively). Additionally, preperitoneal AT also correlated with total mass and tissue (rho = 0.709 for both correlations).

Regarding the RF ultrasound, despite some positive correlations being observed, there were no strong significant correlations between the RF-CSA, RF-AT, RF transversal axis, RF anteroposterior axis, and total body composition parameters determined by DXA (Figure 3B).

### 3.6. Clinical Correlations between Different Body Composition Measurements Determined by DXA and Functional Tests

Some negative correlations were observed between the TUG and body composition parameters determined by DXA, especially with arm LM (rho = −0.686), legs LM (rho = −0.659), and right side LM (rho = −0.650). In contrast, mean handgrip strength positively correlated with arm LM (rho = 0.767), legs LM (rho = 0.764), right side LM (rho = 0.653), and the RSMMI (rho = 0.605; Figure 4).

### 3.7. Clinical Correlations between Body Composition Measurements Using Different Techniques, LVEF and NT-proBNP

Few significant correlations were observed when LVEF was analyzed; specifically, it negatively correlated with arm circumference, ECW, serum albumin, and prealbumin levels; it positively correlated with RF-AT and preperitoneal AT, and all these clinical correlations were weak (rho < 0.4; Figure 5). Additionally, NT-proBNP negatively correlated with mean handgrip strength, arm circumference, body FM, reactance, the RSMMI, the ALT/height^2^ ratio, and serum albumin levels. In contrast, it positively correlated with the gynoid distribution of the FM, total water in %, and ECW (Figure 5).

## 4. Discussion

HF is currently a worldwide problem. It is characterized by significant morbidity and mortality, poor functional capacity, decreased quality of life, and high costs [1]. Current clinical guidelines are focused on treatment optimization and cardiac rehabilitation programs [12,13], but no specific recommendations about nutritional screening, evaluation, or support are available. 

Despite this, there is growing evidence that states an increased incidence of malnutrition in these patients [2], which ranges from 10 to 50%, depending on the stage of disease and according to the used parameters for classifying malnutrition. In this study, we used the current GLIM criteria [5] and the SGA, which offer an overall evaluation of the patient, including a general physical examination and the use of structured clinical parameters to diagnose malnutrition [11]. In our study, according to the GLIM criteria, 54.2% of patients presented with malnutrition; in contrast, using the SGA classification, 39.7% were considered well nourished, 53% were at risk of malnutrition, and 7.3% had malnutrition. The classification of malnutrition according to the GLIM criteria accurately distinguished patients with impaired functionality, lower LM, SMI, and ASMM according to the BIA, lower trunk FM, trunk LM, and fat-free mass according to the DXA, as well as patients with decreased albumin and increased C-RP serum levels. According to SGA, there were significant changes in body composition parameters determined by BIA, muscle ultrasound, and functional tests between well-nourished patients and patients at risk of malnutrition or who had malnutrition, but not when patients at risk were compared with patients with malnutrition, suggesting that both categories should be considered similarly for starting nutritional support and follow up. As previously described, SGA allowed a higher detection of patients at risk of malnutrition than other classification methods [14], including the GLIM criteria [13].

It is well known that BMI is not a reliable marker for evaluating nutritional status in patients with HF [15], not only because it does not evaluate FM and LM distribution but also because patients with HF present with anthropometric changes related to their clinical condition. Particularly, HF is characterized by fluid overload, which collects in the distensible spaces between cells, leading to interstitial edema and macroscopic fluid accumulation in the chest (pleural effusions) or abdomen (ascites) [16]; this abnormal fluid distribution indirectly affects several anthropometric measurements, for example, calf circumference, but fluids could also alter other complex techniques, including BIA or DXA measurements.

According to our study, arm circumference represents a reliable parameter for evaluating malnutrition (and/or sarcopenia) in patients with HF, not only due to its association with the GLIM diagnosis of malnutrition but also with other clinical parameters, including the RF-CSA, determined by ultrasound, and the TUG. The applicability of this easy anthropometric measurement was previously described as a complement to BMI in HF patients [17]. Remarkably, calf circumference was decreased in patients with a risk of malnutrition when the SGA score was used, suggesting that calf circumference could be carefully used in these patients. 

Regarding specific measurements of body composition, decreased lean, fat, and bone mass in patients with HF was described for the first time using DXA by Anker et al. [18]. DXA is considered a gold standard for measuring body composition parameters; unfortunately, is not routinely used due to the required logistics, costs, and low-grade radiation exposure [19]. In this context, novel techniques have emerged, such as safe, easy-to-use, and cheap alternatives for measuring body composition parameters [9,20,21,22]. Among them, electrical bioimpedance analysis (BIA) is probably the most frequently used method. BIA is portable, cheap, easy to use, and reliable, and its results could be altered due to possible errors of standard assumptions on body composition; despite this, novel equipment, which include multifrequency analysis and BIA analysis, improves their measurements and avoids these errors [23].

In this context, previous studies have compared different methods for evaluating body composition; specifically, multifrequency BIA reports similar results for mean fat mass and fat-free mass compared with DXA, but single-frequency devices report significantly higher results for FM and lower values for FFM [24]. In other studies that used only multifrequency devices, DXA consistently gave higher values for FM and lower values for LM than BIA [25], especially in males [26]. It has been reported that both single-frequency and multifrequency BIA methods had wide limits of agreement for FM and fat-free mass compared with DXA in patients with HF [24]. In our study, DXA was compared with BIA, and we observed strong correlations between BCM and body composition parameters determined by DXA, including the RSMMI, and there were also strong correlations between FM and LM using both techniques, especially with the dominant side of the body; additionally, the RSMMI strongly correlated with FFM and ASMM. Finally, PA, but not SPA, correlated with LM but not with LVEF or with NT-proBNP levels. Similarly, a recent study suggested that patients with HF and lower PA values have a 2.11-fold increase in risk of cachexia, but PA values were similar in patients with left ventricular dysfunction and those with alterations in right ventricular function [27]. Based on this, it seems that the value of PA for predicting the clinical evolution of HF is limited.

Despite clinical correlations between FM and LM determined by BIA and DXA being positive and strong, clinical correlations with body composition parameters determined by ultrasound were weaker. Remarkably, RF ultrasound accurately discriminated patients with malnutrition (using GLIM and SGA), but we did not observe clinical correlations with body composition using DXA, even if legs were analyzed separately; additionally, correlations considering abdominal ultrasound were lower and weaker. These findings suggest that RF ultrasound is more valuable for body composition analysis in patients with HF than abdominal ultrasound, but its applicability is limited and requires further validation. Also, when DXA and functional tests were compared, correlations were logical but weak, and this finding could have been due to the fact that patients had recent hospital admissions and were still probably in the recovery phase of the disease.

Interestingly, NT-proBNP showed negative, significant clinical correlations with quality muscle-related parameters determined by different techniques, suggesting that nutritional impact on muscle mass is also reflected in heart functionality. A previous study by our group also revealed these clinical correlations [20], and a recent report in healthy adults described a higher prevalence of increased serum NT-pro BNP in people with low skeletal muscle mass [28].

Regarding biochemical parameters, previous studies have reported prealbumin as a reliable marker for identifying patients with HF at risk of malnutrition [15]; in contrast, in our cohort, there were no statistically significant differences between serum concentrations of albumin/prealbumin and the presence of malnutrition according to GLIM or SGA. Remarkably, prealbumin was negatively correlated with NT-proBNP, suggesting a role in cardiac function; in this line, a previous study described the applicability of the serum cystatin C/prealbumin ratio as a predictive factor for long-term prognosis in patients with chronic heart failure [29]. Additionally, despite the role of serum albumin being initially remarkable in patients with HF [30], further studies have revealed its limitation since liver protein synthesis is reduced in HF [31].

Concerning follow up, in a Spanish study that included 304 patients treated in an HF unit, mortality in patients with malnutrition (classified using the Mini Nutritional Assessment screening test) was higher (68.9%) compared to those who were at risk of malnutrition (33.3%) or non-malnourished patients (15.2%; *p* < 0.001); in that cohort, malnutrition was an independent predictor of mortality [3]. In another study performed in 130 ambulatory patients, the presence of malnutrition according to the same screening tool was an independent predictor of muscle wasting and mortality [32], similar to our study. In contrast, after a mean of 13 months of follow up, we did not observe clinical associations between the diagnosis of malnutrition and mortality (only age- and sex-adjusted baseline LVEF, percentage of body weight loss, and the arm circumference tended to correlate) or new HF hospital admissions (only preperitoneal AT was significantly associated); differences might be explained by the time of follow up (28 months in the first study) and different classification systems. Despite this, results regarding mortality and malnutrition in HF are contradictory; specifically, two metanalyses show contradictory results [33,34].

This study has some limitations, including the number of participants. DXA was only performed in a small group of patients and, finally, the follow up was short and did not include nutritional evaluation. In contrast, this study has several strengths. First of all, a systematic nutritional evaluation was performed, including anthropometric, echography, functional, technical, and biochemical parameters. Patients were evaluated after a recent hospital admission due to HF; thus, clinically affected patients and, finally, several methods for diagnosing malnutrition and assessing body composition were compared.

## 5. Conclusions

Taken together, our results reveal that SGA allows higher detection of patients at risk of malnutrition compared with GLIM criteria, allowing an earlier nutritional intervention. Thus, it could play a significant role in preventing malnutrition (and its consequences) in these patients. Despite this, for an accurate diagnosis, GLIM criteria should be used. BIA and DXA showed significant correlations for evaluating FM and LM in patients with HF, despite the fact that they are not interchangeable; both methods are useful and have their applicability in these patients, depending on their availability. If it is not possible to determine body composition parameters, arm circumference provides general but relevant information on the nutritional status of the patient, even when functional tests could be altered to the clinical condition of the patient (regarding heart functionality). Finally, a larger follow up should be performed to determine the specific role of nutritional ultrasound in patients with HF.

## Figures and Tables

**Figure 1 nutrients-16-01535-f001:**
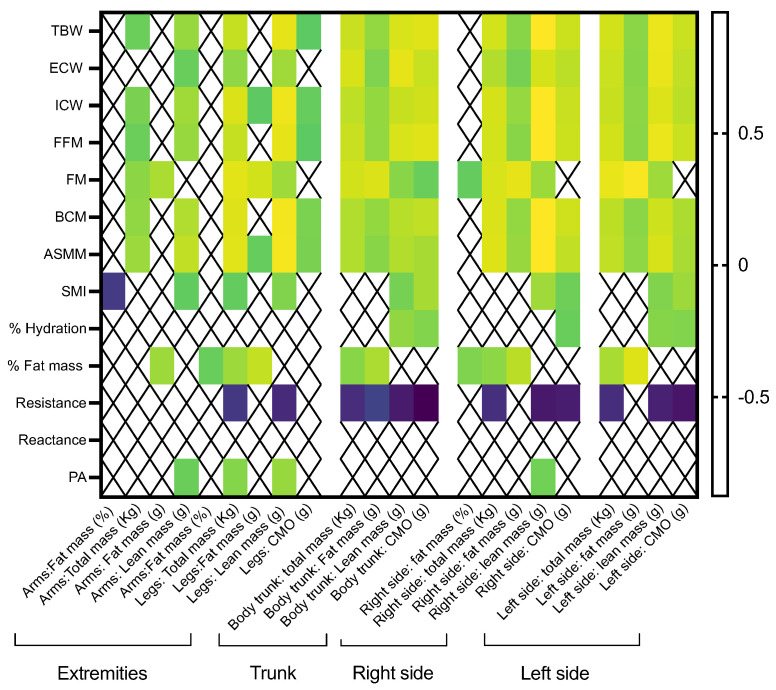
Significant correlations between body composition parameters determined by BIA (*y*-axis) and DXA (extremities, trunk, and different sides, *x*-axis). Only statistically significant correlations are presented. Legend: TBW: total body water; ECW: extracellular body water; ICW: intracellular body water; FFM: fat-free mass; FM: fat mass; BCM: body cell mass, ASMM: appendicular muscle mass; SMI: skeletal mass index; PA: phase angle; CMO: bone cell mass.

**Figure 2 nutrients-16-01535-f002:**
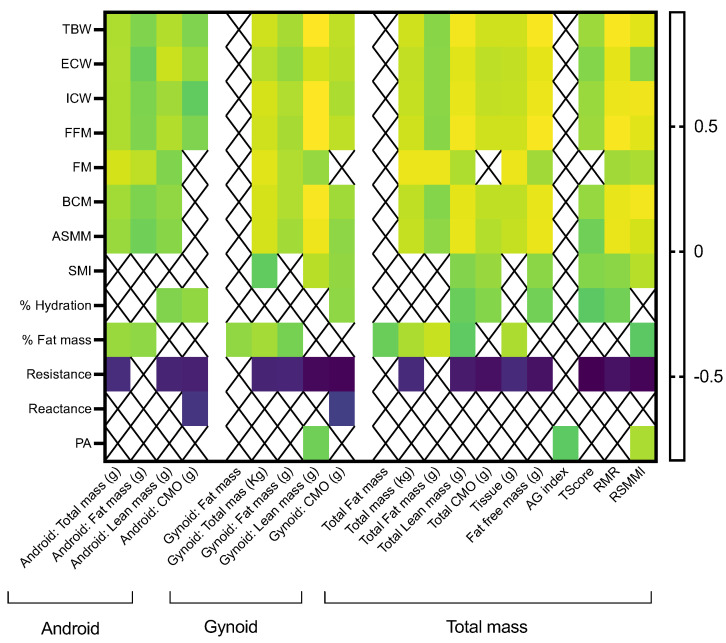
Significant correlations between body composition parameters determined by BIA (*y*-axis) and DXA (total measurements, *x*-axis). Only statistically significant correlations are presented. Legend: TBW: total body water; ECW: extracellular body water; ICW: intracellular body water; FFM: fat-free mass; FM: fat mass; BCM: body cell mass, ASMM: appendicular muscle mass; SMI: skeletal mass index; PA: phase angle; CMO: bone cell mass.

**Figure 3 nutrients-16-01535-f003:**
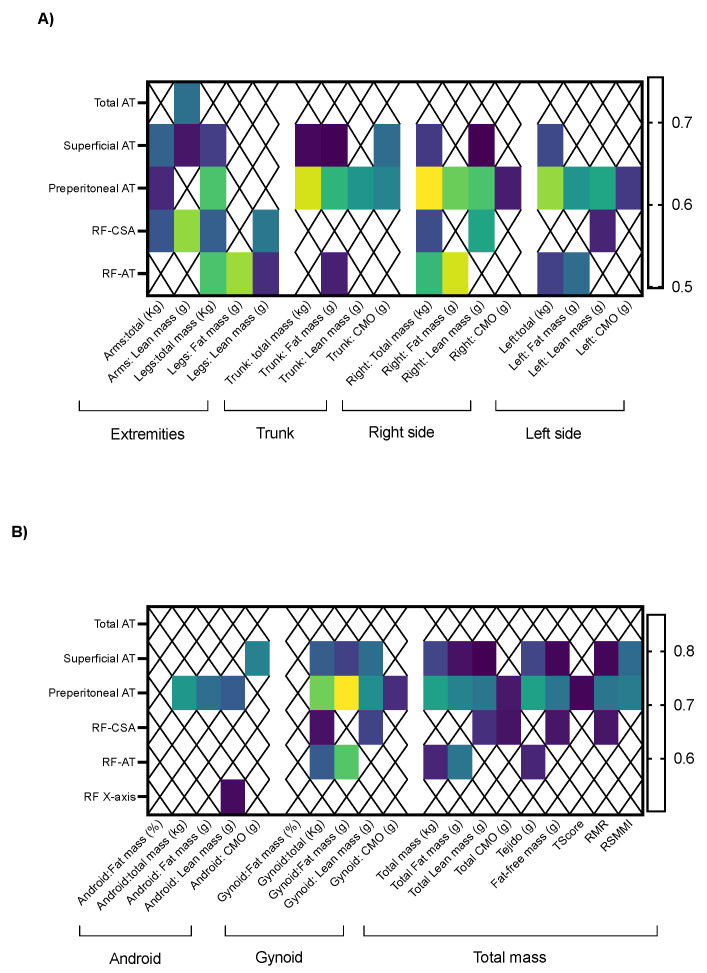
Significant correlations between body composition parameters determined by nutritional ultrasound and DXA: (**A**) extremities, trunk, and different sides; (**B**) total body composition parameters. Legend: AT: adipose tissue; CMO: bone cell mass; RMR: resting metabolic rate; RSMMI: relative skeletal muscle mass index.

**Figure 4 nutrients-16-01535-f004:**
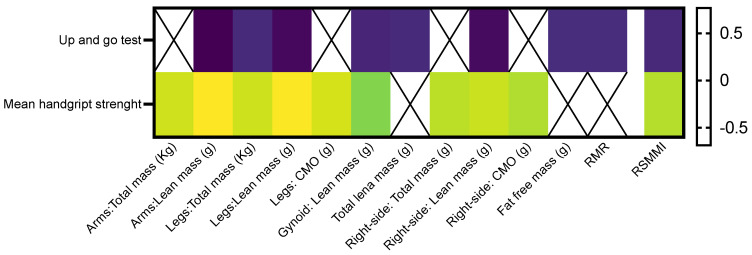
Significant correlations between body composition parameters determined by DXA and functional nutritional parameters. Legend: CMO: bone cell mass; RMR: resting metabolic rate; RSMMI: relative skeletal muscle mass index.

**Figure 5 nutrients-16-01535-f005:**
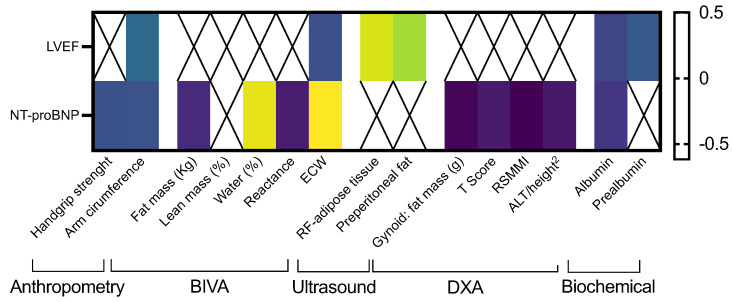
Significant correlations between body composition parameters determined by different techniques, LVEF and serum NT-proBNP. Legend: ECW: extracellular water; RSMMI: relative skeletal muscle mass index; LVEF: left ventricular ejection fraction.

**Table 1 nutrients-16-01535-t001:** Baseline clinical characteristics of the patients. Comparison between groups based on the presence of malnutrition according to the GLIM criteria.

Characteristics	Total (*n* = 83)	No Malnutrition (*n* = 38)	Malnutrition (*n* = 45)	*p*
Sex (♂/♀)	75.9%/24.1% (63/20)	73.7/26.3 (28/10)	22.2/77.8 (35/10)	0.43
Age (years)	65.7 (27–81)	65.5 (55–75)	71 (59–79)	0.15
Tobacco exposure	21.7 (18/83)	21.1 (8/38)	22.2 (10/45)	0.56
Type 2 diabetes	44.4 (45/81)	41.7 (15/36)	46.7 (21/45)	0.41
Previous ischaemic cardiomyopathy	48.1 (38/81)	52.8 (19/36)	44.4 (20/45)	0.30
Ejection fraction	49 (32–65)	50 (32–63)	48.5 (33–66)	0.84
NT-proBNP	2931 (821–5930)	2257 (780–5146)	3185 (1086–6675)	0.44
**Symptoms**				
Weight loss (6 months)	59 (49/83)	36.8 (14/38)	77.8 (35/45)	<0.01
Weight loss (Kg)	2.0 (0–5.3)	2.3 ± 2.9	2.6 ± 2.6	0.56
**Preferred food intake**				0.001
Liquid food	33.7 (28/83)	52.6 (20/38)	17.8 (8/45)	
Soft food	24.1 (20/83)	7.9 (3/38)	37.8 (17/45)	
Normal texture	42.2 (35/83)	39.5 (15/38)	44.4 (20/45)	
**Gastrointestinal symptoms**	20.5 (1/83)	18.4 (7/38)	22.2 (10/45)	0.44
Abdominal pain	6 (5/83)	5.3 (2/38)	6.7 (3/45)	0.58
Nauseous/vomits	13.3 (11/83)	10.5 (4/38)	15.6 (7/45)	0.37
Diarrhea	4.8 (4/83)	2.6 (1/38)	6.7 (3/45)	0.38
**Physical activity**				
Intense	0	0	0	-
Moderate	22.9 (19/83)	36.8 (14/38)	11.1 (5/45)	0.006
Hospital admissions due to HF (%)	93.8 (76/81)	94.4 (34/36)	93.3 (42/45)	0.60
Number of hospital admissions	2 (1–4)	2 (1–3)	2 (1–4)	0.70
Mortality	44.4 (36/81)	2.8/1/36)	11.1 (5/45)	0.16

**Legend**: Continuous data are expressed as median with IQR, categorical data are expressed as a percentage, and the absolute numbers are expressed in brackets.

**Table 2 nutrients-16-01535-t002:** Baseline clinical characteristics of the patients. Comparison between groups based on the presence of malnutrition or risk of malnutrition according to the subjective global assessment (SGA).

Characteristics	Total (*n* = 83)	Well Nourished (*n* = 33)	At risk of Malnutrition (*n* = 44)	Malnutrition (*n* = 6)	*p*
Sex % (♂/♀)	75.9/24.1 (63/20)	87.9/12.1 (29/4)	68.2/38.8 (30/14)	66.7/33.3 (4/2)	0.12
Age (years)	65.7 (27–81)	61 (54–70)	70 (61–79)	78.5 (75–81)	0.64
Tobacco exposure	21.7 (18/83)	27.3 (9/33)	18.2 (8/44)	16.1 (1/6)	0.60
Type 2 diabetes	44.4 (45/81)	33.3 (11/33)	52.4 (22/42)	50 (3/6)	0.14
Previous ischaemic cardiomyopathy	48.1 (38/81)	48.5 (16/33)	50 (21/42)	33.3 (2/6)	0.72
Ejection fraction	49 (32–65)	44.5 (29–54)	56 (35–66)	66 (41–71)	0.20
NT-proBNP	2931 (821–5930)	2170 (779–6600)	2967 (1043–4633)	4000 (1071–6976)	0.65
**Symptoms**					
Weight loss (6 months)	59 (49/83)	42.4 (14/33)	70.5 (31/44)	66.7 (4/6)	0.04
Weight loss (Kg)	2.0 (0–5.3)	0 (0–2.3)	3 (0–6)	6.5 (2–12)	0.009
**Food intake**					0.007
Liquid food	33.7 (28/83)	42.4 (14/33)	31.8 (14/44)	0	
Soft food	24.1 (20/83)	6.1 (2/33)	31.8 (14/44)	66.7 (4/6)	
Normal texture	42.2 (35/83)	51.1 (17/33)	36.4 (16/44)	33.3 (2/6)	
**Gastrointestinal symptoms**	20.5 (1/83)	18.2 (6/33)	20.5 (9/44)	33.3 (2/6)	0.67
Abdominal pain	6 (5/83)	3 (1/33)	6.8 (3/44)	16.7 (1/6)	0.41
Nauseous/vomits	13.3 (11/83)	9.1 (3/33)	15.9 (7/44)	16.7 (1/6)	0.66
Diarrhea	4.8 (4/83)	6.1 (2/33)	2.3 (1/44)	16.7 (1/6)	0.28
**Physical activity**					
Intense	0	0		0	
Moderate	22.9 (19/83)	45.5 (15/33)	6.8 (3/44)	16.7 (1/6)	<0.001
Hospital admissions due to HF (%)	93.8 (76/81)	90.9 (30/33)	97.6 (41/42)	83.3 (5/6)	0.80
Number of hospital admissions	2 (1–4)	1 (1–1)	1 (1–1)	1 (1–1)	0.42
Mortality	44.4 (36/81)	3 (1/33)	9.5 (4/42)	16.7 (1/6)	0.16

**Legend**: Continuous data are expressed as median with IQR, categorical data are expressed as a percentage, and the absolute numbers are expressed in brackets.

**Table 3 nutrients-16-01535-t003:** Morphofunctional assessment of the nutritional status according to the GLIM criteria or SGA.

		GLIM			SGA					
Characteristics	Total (*n* = 83)	No Malnutrition (*n* = 38)	Malnutrition (*n* = 45)	*p*1	Well Nourished (*n* = 33)	At Risk of Malnutrition (*n* = 44)	Malnutrition (*n* = 6)	*p*2	*p*3	*p*4
Body weight	77 (66–83)	82.5 (70.2–91.4)	72 (64–81.5)	0.01	82.5 (72–90)	71 (61–83)	58.5 (50.5–10.6)	0.003	0.003	0.11
Percentage of body weight loss	2.3 (−1.2–7.1)	0.6 (−3–2.7)	4 (0.6–11.3)	0.001	0.9 (−2.8–3.3)	3.2 (0–8)	8 (0.1–18)	0.08	0.14	0.37
**BIA**										
BMI (kg/m^2^)	27.5 (23.9–30.6)	28.7 (25.9–31)	26.4 (23.4–29.9)	0.09	28.9 (26–31.8)	26.7 (23.3–35.3)	22.5 (18.3–27.6)	0.04	0.04	0.18
BCMe	32.2 (24.6–38.3)	34.8 (25.1–39.7)	28.7 (24.2–34.8)	0.14	36.5 (28.1–41.1)	30.1 (23–35.3)	25.9 (24–31.2)	0.005	0.01	0.46
ECMe	20.1 (19.8–22.8)	21.5 (19.9–22.9)	20.4 (19.9–22.4)	0.72	22.4 (21.5–23.7)	20 (18.6–21.5)	20.2 (17.3–20.5)	0.02	0.07	0.84
Fat mass (%)	23.3 (16.1–28.5)	25 (18.7–29.7)	22 (14.8–26.3)	0.21	25.1 (19.7–30.6)	22.4 (15.5–26.2)	17.1 (6.8–24.5)	0.14	0.16	0.32
Fat mass (kg)	20.2 (14.6–25.8)	22 (15.8–23.6)	18.3 (14.8–26.3)	0.96	22.9 (20.2–27.2)	17.2 (11.3–23.3)	19.4 (13.7–23.7)	0.02	0.48	0.69
Lean mass (%)	71.2 (65.1–74.2)	71.2 (67.4–73.4)	58.1 (50.3–65.8)	0.94	69.7 (65.1–72.4)	73 (66–76)	70 (66.3–74.6)	0.18	0.92	0.69
Lean mass (kg)	53.3 (48.7–61.5)	58.1 (50.3–65.8)	52.3 (45.7–59.3)	0.02	60.2 (52.3–64.5)	52.1 (45–59.1)	47.6 (41.1–50.3)	0.008	0.03	0.12
Water (%)	52.6 (48.2 -55.5)	53.1 (48.7–54.2)	52.4 (47.4–57)	0.92	51.3 (49.1 -53.9)	54.2 (48.2–56.9)	51 (48.8–54.8)	0.25	0.96	0.77
Water (kg)	39.9 (34.1–47)	42.8 (35.2–49)	38.1 (33.4–43.2)	0.03	44.2 (38.1–49)	38 (33–43)	35 (30.1–36.9)	0.006	0.003	0.13
Bone mass (kg)	2.8 (2.7–3.2)	3 (2.8–3.3)	2.8 (2.7–3.2)	0.40	3.2 (3–3.3)	2.8 (2.6–3)	2.7 (2.4–2.8)	0.01	0.05	0.48
Phase angle	5.35 (4.4–6.1)	5.6 (4.5–6.4)	5 (4.3–6.1)	0.19	6 (4.6–6.3)	5 (4.4–6)	4.5 (3.8–5.7)	0.19	0.31	0.60
Resistance	542 (445–595)	482 (430–586)	558 (493–594)	0.21	499 (446–594)	546 (439–614)	551 (549–574)	0.63	0.36	0.69
Reactance	53.5 (43.8–58.3)	52.5 (47.5–58)	55 (43–61)	0.88	56 (51.3–61.2)	52 (41.5–57.5)	48 (43.5–55)	0.10	0.30	0.94
ECW (kg)	18.2 (16.2–21.1)	20.4 (17–22)	17.5 (15.6–20.1)	0.29	18.5 (17.4–21.6)	19.6 (15.3–21)	16.3 (15.8–17.4)	0.93	0.20	0.47
ICW (kg)	20.2 (17.7–24)	22.6 (18.7–27.8)	19 (15.4–21.6)	0.04	22.4 (20.4–26.8)	19.6 (15.3–23)	17.9 (15.2–18)	0.04	0.08	0.28
ASMM (kg)	21 (17.2–25.1)	22.2 (18.8–26.9)	19.9 (16.6–21.2)	0.03	22.4 (20.4–26.8)	20.5 (15.5–22.9)	17.2 (15.4–21.6)	0.06	0.03	0.22
SMI (cm^2^/m^2^)	8.5 (7.9–10.5)	10.2 (8.2–10.9)	8.3 (7.3–9.9)	0.05	9.9 (8.5–10.6)	8.4 (7.4–10.5)	8.1 (7.6–8.8)	0.14	0.11	0.66
**Anthropometric evaluation**										
Abdominal circumference	101 (93.5–109.5)	98 (96.5–99.5)	101 (92–113)	0.73	113 (107–125)	97.5 (90–102)	97.5 (90–102)	0.06	0.06	0.27
Arm circumference	28.7 (25.6–32)	30 (28–32)	27 (25–31)	0.02	31 (29–33)	28 (25–31)	25 (22–27)	<0.001	0.01	0.23
Calf circumference	36 (32–39)	37 (34–39)	36 (32–38)	0.21	37 (35–39)	34 (32–39)	31 (29–35)	0.02	0.002	0.04
**RF Muscle ultrasound**										
Adipose tissue (cm)	3.3 (0.7–7)	4.92 (0.8–8.2)	2.1 (0.6–6)	0.09	2 (0.7–6)	4.8 (0.8–8.3)	1.9 (0.8–3.7)	0.33	0.56	0.22
CSA (cm^2^)	3.7 (2.7–4.6)	4.1 (3.3–4.6)	3.4 (2.5–4.1)	0.04	4 (2.9–4.8)	3.5 (2.5–4.3)	3.5 (3.3–3.9)	0.10	0.67	0.74
Circumference (cm)	8.7 (7.5–10)	9.7 (8.1–11)	8.3 (6.8–9.6)	0.08	9.4 (7.7–10.3)	8.6 (7.1–9.7)	7.5 (7.4–7.7)	0.34	0.15	0.36
AP axis (cm)	1.2 (0.88–1.3)	1.3 (1–1.4)	1 (0.8–1.3)	0.01	1.3 (1.1–1.4)	1.1 (0.8–1.3)	1 (0.9–1.4)	0.02	0.69	0.36
Transversal axis (cm)	3.9 (3.3–4.4)	4.2 (3.5–4.5)	3.7 (3.1–4.1)	0.01	4.2 (3.4–4.5)	3.8 (3.4–4.3)	2.5 (2.4–2.9)	0.09	0.01	0.02
**Abdominal ultrasound**										
Subcutaneous adipose tissue (cm)	7.6 (1.7–13.2)	9.6 (2.3–13.8)	3 (1.5–13.1)	0.06	6.2 (1.7–12)	9.3 (1.7–14.6)	1.9 (1.1–5.2)	0.51	0.13	0.06
Preperitoneal fat (cm)	2.4 (0.6–5.5)	3.3 (0.8–6.8)	1.6 (0.53–4)	0.11	1.7 (0.6–4.8)	2.9 (0.7–6.8)	1.3 (0.4–3.4)	0.73	0.21	0.11
**Functional evaluation**										
Handgrip strenght (dominant arm, kg)	30 (22–39)	34.8 (26.5–42.2)	23 (19.6–34.6)	0.002	40 (25–45)	28 (20–33)	18 (16–29)	<0.001	0.03	0.27
Up-and-go test (seconds)	17.3 (12.6–20.8)	15 (10–18)	19 (15–25)	<0.001	15.6 (9.8–18.3)	17.3 (14–25)	21 (19–22)	0.04	0.03	0.29

**Legend:** Continuous data are expressed as median with interquartile range (IQR), categorical data are expressed in percentage, and the absolute numbers are expressed in brackets. BIA: bioimpedance analysis; BMI: body mass index; BCMe: body cellular mass; ECMe: extracellular mass; ECW: extracellular water; ICW: intracellular water; ASMM: appendicular skeletal muscle mass; SMI: skeletal muscle mass index; RF: rectus femoris; AP: anteroposterior; CSA: cross-sectional area. *p*1 refers to the comparison between patients with and without malnutrition according to the GLIM criteria; *p*2 refers to the comparison between well-nourished patients and patients at risk of malnutrition; *p*3 refers to the comparison between well-nourished patients and patients with malnutrition; *p*4 refers to the comparison between patients at risk of malnutrition and malnourished patients.

**Table 4 nutrients-16-01535-t004:** Body composition parameters determined by DEXA according to the GLIM criteria or SGA.

		GLIM					
Characteristics	Total (*n* = 16)	No Malnutrition (*n* = 8)	Malnutrition (*n* = 8)	*p*1	Well Nourished (*n* = 8)	At Risk of Malnutrition and Malnutrition (*n* = 8)	*p*2
**DEXA**							
Arms total mass (kg)	9.0 (7.9–10.3)	9.8 (9–10.5)	8.2 (7.6–9)	0.20	8.5 (6.3–10.2)	9.6 (7.8–10.3)	0.90
Arm fat mass (%)	33.9 (30.7–37.1)	32 (30.7–37.3)	35.7 (31–37)	0.72	33.9 (32.2–35.6)	31 (29–37.9)	0.86
Arm fat mass (kg)	2.7 (2.4–3.1)	29.1 (27.5–34.4)	23 (22.4–24.5)	0.08	2.8 (2.7–3.1)	2.7 (2–3.2)	0.77
Arm lean mass (kg)	5.8 (5.2–6.5)	61 (56.4–66.6)	53.1 (46.8–59.8)	0.22	5.6 (5.2–6.5)	6.2 (5–6.6)	0.95
Leg total mass (kg)	25.9 (22.1–30.7)	30.3 (25.3–32)	22.6 (21.1–27.2)	0.08	27.2 (23.2–32)	26.3 (23.9–30.3)	0.86
Leg fat mass (%)	3.6 (2.8–3.6)	31.5 (27.7–33.5)	33.3 (28.6–38.4)	0.60	33.2 (28.2–36)	26.3 (23.9–30.3)	0.95
Leg fat mass (g)	8 (6.8–9.5)	8.1 (7.6–9.8)	7.3 (5.7–9)	0.28	8.4 (7–10.5)	7.9 (7.3–8.6)	0.86
Leg lean mass (g)	17 (15.2–20.2)	18.9 (16.3–21.1)	15.3 (13.5–18.1)	0.11	17.4 (15.7–21.1)	17.7 (15.9–19)	0.77
Trunk total mass (kg)	42.6 (40–53.3)	50.9 (44.1–53.9)	37.5 (32.9–42.2)	0.01	41.3 (38.7–49.6)	44.8 (42.6–54.2)	0.15
Trunk fat mass (%)	39.2 (36–42.7)	39.6 (37.8–42.7)	38.4 (34.5–41.2)	0.57	38.6 (36.3–41.2)	40.1 (36.7–42.8)	0.95
Trunk fat mass (g)	16.5 (13.1–19.6)	18.6 (17–22.4)	13.1 (12.5–15.4)	0.02	16.4 (14.3–18.3)	17.5 (14.2–23.6)	0.69
Trunk lean mass (g)	25.6 (22.3–29.3)	28.6 (26.5–32)	22 (20.3–25.1)	0.02	24.2 (22.3–27.1)	27.9 (25.5–30.6)	0.28
Android total mass (kg)	7.2 (6.4–9)	8.2 (7.2–9.5)	6.3 (5–7)	0.04	7.2 (6.2–7.9)	7.4 (6.8–9.8)	0.53
Gynoid total mass (kg)	11.8 (10.4–14.1)	13.7 (12.6–15.4)	10.2 (9.2–11.2)	0.03	11.3 (10.4–14.1)	12.9 (11.3–14.5)	0.86
Total fat mass (%)	35.9 (34–37)	100 (82.7–101)	75.2 (68.5–80)	0.96	35.8 (34.7–36.9)	35.8 (34.1–37.5)	0.46
Total body mass (kg)	81.4 (75.5–100.3)	35.6 (34–37.3)	36 (34–36.9)	0.05	78.3 (75.5–99.2)	85 (81.2–101.7)	0.61
Total fat mass (g)	27.4 (25.9–33.3)	32.3 (27.9–35.2)	25.4 (22.7–27.1)	0.02	27.1 (26.7–32.2)	28.1 (25.7–35.7)	0.61
Total lean mass (g)	51.1 (46.5–59.1)	57.7 (52–62.3)	46.5 (44.5–50.3)	0.02	48.7 (46.5–56.6)	54 (50.6–60.5)	0.61
Total CMO (g)	28.7 (25.8–32.9)	31.9 (29.8–33.4)	25.9 (25.4–27.7)	0.13	2.9 (2.6–3.2)	3.1 (2.6–3.4)	0.90
Right side fat mass (%)	13.6 (12.5–16.8)	49.1 (39.8–49.7)	37.3 (34.3–37)	0.80	36 (34.6–37)	35.3 (34–37.8)	0.53
Right side total mass (g)	39.4 (37.4–49.2)	35.8 (34.2–37.6)	35.9 (33.5–37)	0.05	38.3 (37.4–49.2)	41.6 (39.1–49.8)	0.86
Right side fat mass (g)	13.6 (12.5–16.8)	16 (13.5–17.4)	12.3 (11.1–13.6)	0.04	13.6 (12.9–15.8)	13.7 (12.4–17.5)	0.77
Right side lean mass (g)	24.6 (23–28.8)	28 (25.1–30.1)	23.1 (22.4–24.5)	0.02	23.5 (23.3–28)	27.8 (26.2–30.8)	0.39
Left side fat mass (%)	36.3 (35–37.6)	49.8 (43–51.3)	37.1 (34.2–42.2)	0.72	1.4 (1.3–1.6)	1.4 (1.3–1.8)	0.39
Left side total mass (g)	42.6 (37.1–50.1)	36 (35–40)	36.3 (34.2–36.8)	0.03	25.3 (23.3–28.4)	27.8 (26.2–30.8)	0.86
Left side fat mass (g)	14.4 (12.9–16.5)	16.3 (14.4–17.8)	12.6 (11.6–13.7)	0.02	1.4 (1.3–1.6)	1.6 (1.3–1.7)	0.33
Left side lean mass (g)	27.7 (23.3–30)	29.5 (26.9–32.2)	23.2 (22.1–26.3)	0.01	75.5 (72.9–96)	81.5 (78.4–98.6)	0.46
Total tissue (g)	78.5 (72.9–96.9)	96.2 (79.6–97.8)	72.6 (65.8–77.4)	0.05	51.4 (49.2–59.8)	56.7 (53.4–64)	0.61
Fat-free mass (g)	54 (49.2–65.5)	60.7 (55.4–66)	50 (47.1–53)	0.01	1.2 (1.1–1.3)	1.1 (1–1.2)	0.18
A/G index	1.2 (1.1–1.2)	1.1 (1.1–1.2)	1.2 (0.9–1.2)	0.96	0.5 (−0.2–2)	0.4 (−0.4–1.8)	0.96
T score	0.3 (−0.8–2)	1.8 (0.3–2.3)	−0.5 (−0.8–0.3)	0.09	8.3 (7.7–8.9)	8.2 (7.8–8.5)	0.61
RSMMI	8.1 (7.4–8.6)	8.5 (8.1–8.8)	7.6 (6.6–8.2)	0.06	22.9 (20.9–27.8)	23.8 (21.9–24.8)	0.53
ALM	23.1 (20.8–25.9)	24 (22.2–28.2)	20.7 (18.3–24.3)	0.11	8.3 (7.6–8.9)	8.1 (7.6–8.9)	0.87
ALM/height^2^	8.1 (7.3–8.7)	8.5 (8–8.8)	7.5 (6.4–8.2)	0.11	22.9 (20.9–27.8)	23.8 (21.9–24.8)	0.61
ALM/height^2^ < 7 (%)	20	0	20	0.48	8.3 (7.6–8.9)	8.1 (7.6–8.9)	0.11

**Legend:** Continuous data are expressed as median with IQR. DEXA: dual-energy X-ray absorptiometry; CMO: bone mineral content; A/G: android/gynoid; RSMMI: relative skeletal muscle mass index; ALM: appendicular lean mass.

**Table 5 nutrients-16-01535-t005:** Biochemical analysis (nutritional parameters) according to the GLIM criteria or SGA.

		GLIM			SGA					
Characteristics	Baseline (*n* = 38)	Malnutrition (*n* = 19)	No Malnutrition (*n* = 15)	p1	Well Nourished (*n* = 33)	At risk of Malnutrition (*n* = 44)	Malnutrition (*n* = 6)	*p*2	*p*3	*p*4
Hemoglobin	13.8 (13–15.3)	13.9 (13.1–15.4)	13.7 (12.5–15)	0.43	15.1 (13.3–15.4)	13.6 (13 -15)	12.9 (12.4–13.1)	0.26	0.26	0.19
Lymphocytes	2400 (1600–3670)	2200 (1400–1730)	38125 (1675–1625)	0.87	835 (1570–1777)	2200 (1800–1495)	625 (95–1707)	0.62	0.62	0.65
Albumin (g/dL)	4.3 (4–4.6)	4.4 (4.1 -4.7)	4.2 (3.8–4.6)	0.07	4.3 (4.1–4.9)	4.3 (3.8 -4.6)	4.2 (3.9–4.6)	0.13	0.13	0.96
Prealbumin (mg/dl)	23 (18–27)	24 (19–28)	22 (17–27)	0.28	24 (19–28)	23 (17–27)	22 (19–26)	0.15	10.15	0.85
Ferritin (mg/dL)	106 (35–176)	103 (54–130)	120 (39–182)	0.48	108 (63–179)	106 (34–145)	30 (29–107)	0.72	0.72	0.77
Transferrin (mg/dL)	246 (218–262)	245 (225–250)	252 (215–279)	0.43	244 (225–250)	252 (220–300)	244 (228–273)	0.82	0.27	0.74
Total cholesterol (mg/dL)	145 (116–177)	154 (134–177)	135 (114–175)	0.31	145 (116–177)	154 (134–177)	125 (114–175)	0.27	0.26	0.21
HDL cholesterol (mg/dL)	46 (41–53)	47 (45 -52)	45 (37–54)	0.28	45 (44–53)	45 (38 -55)	49 (48–49)	0.50	0.50	0.53
LDL cholesterol (mg/dL)	86 (65–115)	100 (83–120)	139 (42–103)	0.02	99 (73–127)	86 (53–103)	69 (48–72)	0.22	0.22	0.26
Triglycerides (mg/dL)	136 (101–174)	133 (88–170)	139 (102–182)	0.58	139 (103–228)	140 (86–167)	129 (116–159)	0.52	0.52	0.74
C-RP (mg/L)	4 (1.2–13.9))	3.7 (0.9–6.7)	6.8 (1.7–26)	0.05	4 (1–8.6)	4.2 (1.6–20.5)	2.2 (0.8–6.4)	0.35	0.35	0.34
NT-proBNP (pg/mL)	1854 (1080–4364)	1348 (802–2751)	1952 (1179–5405)	0.31	1197 (669–1686)	2196 (1566–6369)	4784 (4700–5405)	0.04	0.04	0.83
Vitamin D (ng/dL)	17.6 (11–26)	23 (17–26)	13 (11–21)	0.02	17.6 (13–26)	18 (11–22)	13 (11–23)	0.73	0.73	0.95

**Legend:** Continuous data are expressed as median with IQR. *p*1 refers to the comparison between patients with and without malnutrition according to the GLIM criteria; *p*2 refers to the comparison between well-nourished patients and patients at risk of malnutrition; *p*3 refers to the comparison between well-nourished patients and patients with malnutrition; *p*4 refers to the comparison between patients at risk of malnutrition and malnourished patients.

**Table 6 nutrients-16-01535-t006:** Multivariate logistic regression for mortality and new hospital admissions in patients with HF that received nutritional support after adjusting by age and sex.

Variable		OR	CI	*p*
Mortality	Baseline LVEF	1.06	0.99–1.13	0.06
	Arm circumference	0.82	0.67–1.0	0.06
	Percentage of body weight loss	1.10	0.99–1.23	0.06
	Control NT-proBNP	1.00	1.00–1.10	0.02
New hospital admissions	Preperitoneal fat	0.73	0.55–0.95	0.01

## Data Availability

The original contributions presented in the study are included in the article, further inquiries can be directed to the corresponding author.

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
