# Peer review of "Differences in the Evaluation of Malnutrition and Body Composition Using Bioelectrical Impedance Analysis, Nutritional Ultrasound, and Dual-Energy X-ray Absorptiometry in Patients with Heart Failure"

_nutrients, 2024, doi:10.3390/nu16101535_

Round 1

Reviewer 1 Report

Comments and Suggestions for Authors

Thank you for the opportunity to review the article on the study on differences in malnutrition and body mass determined by various methods. In the study a nutritional assessment was carried out among patients with heart failure, including anthropometric, echographic, functional, technical and biochemical parameters. Patients who had recently been admitted to hospital due to HF, i.e. patients with clinical disease, were assessed, and several methods for diagnosing malnutrition and assessing body composition were compared. 

Due to the increasing evidence indicating an increased incidence of malnutrition in these patients, it is important to assess the nutritional status. The specific measurement of lean, fat and bone mass using DXA is considered the gold standard for measuring body composition parameters, but is not routinely used primarily due to costs and exposure to low-energy radiation. Bioelectrical impedance analysis (BIA) used in the study is a new technique, safe, easy to use and cheap alternative to measuring body composition parameters. Innovative equipment including multi-frequency analysis and BIVA analysis improves measurements and avoids possible errors resulting from standard assumptions about body composition. 

In the authors' study, it was accurately used to determine nutritional status parameters according to the current GLIM criteria. The SGA scale was also used, which allows for an overall assessment of the patient, includes a general physical examination and uses structured clinical parameters for diagnosis. The use of various methods to identify malnutrition is noteworthy, including taking into account the diagnostic criteria for malnutrition established by nutrition experts and increasingly disseminated around the world, such as the GLIM criteria. An introduction related to the subject of the study, material and methods explain the research process. The results are presented in a fairly orderly manner, including tables and figures. The discussion explains and justifies the obtained results and refers to publications by other authors of works on similar topics. Limitations of the study presented reasonably.

Based on the authors' results: "According to the GLIM criteria, 54.2% of patients presented with malnutrition, in contrast, using the SGA classification, 39.7% were considered well-nourished, 53% at risk of malnutrition and 7.3% with malnutrition" and the conclusion: "Taken together, our results reveal that SGA allows higher detection of patients at risk of malnutrition compared with GLIM criteria, allowing an earlier nutritional intervention." I see some inaccuracy in this statement, because the GLIM showed a higher percentage of malnourished patients, which indicates a justified and absolute inclusion of nutritional treatment, however, based on SGA only risk of malnutrition was detected. It is recommended to interpret/explain this statement why the SGA scale is better.

Author Response

We sincerely thank the Editor for the interest in our patient cohort and results. Following your suggestions, we re-evaluated our manuscript and focused on the comments of the Reviewers. We sincerely thank the Reviewers for their constructive comments, which we found very helpful towards improving the quality of our study. Accordingly, specific changes have been made in the manuscript, based on these comments, as it is described in detail below in a point-by-point description of the changes introduced, and on how Reviewer’s concerns were addressed. Changes within the manuscript are indicated in red.

Reviewer: Thank you for the opportunity to review the article on the study on differences in malnutrition and body mass determined by various methods. In the study a nutritional assessment was carried out among patients with heart failure, including anthropometric, echographic, functional, technical and biochemical parameters. Patients who had recently been admitted to hospital due to HF, i.e. patients with clinical disease, were assessed, and several methods for diagnosing malnutrition and assessing body composition were compared. 

Due to the increasing evidence indicating an increased incidence of malnutrition in these patients, it is important to assess the nutritional status. The specific measurement of lean, fat and bone mass using DXA is considered the gold standard for measuring body composition parameters, but is not routinely used primarily due to costs and exposure to low-energy radiation. Bioelectrical impedance analysis (BIA) used in the study is a new technique, safe, easy to use and cheap alternative to measuring body composition parameters. Innovative equipment including multi-frequency analysis and BIVA analysis improves measurements and avoids possible errors resulting from standard assumptions about body composition. 

In the authors' study, it was accurately used to determine nutritional status parameters according to the current GLIM criteria. The SGA scale was also used, which allows for an overall assessment of the patient, includes a general physical examination and uses structured clinical parameters for diagnosis. The use of various methods to identify malnutrition is noteworthy, including taking into account the diagnostic criteria for malnutrition established by nutrition experts and increasingly disseminated around the world, such as the GLIM criteria. An introduction related to the subject of the study, material and methods explain the research process. The results are presented in a fairly orderly manner, including tables and figures. The discussion explains and justifies the obtained results and refers to publications by other authors of works on similar topics. Limitations of the study presented reasonably.

Based on the authors' results: "According to the GLIM criteria, 54.2% of patients presented with malnutrition, in contrast, using the SGA classification, 39.7% were considered well-nourished, 53% at risk of malnutrition and 7.3% with malnutrition" and the conclusion: "Taken together, our results reveal that SGA allows higher detection of patients at risk of malnutrition compared with GLIM criteria, allowing an earlier nutritional intervention." I see some inaccuracy in this statement, because the GLIM showed a higher percentage of malnourished patients, which indicates a justified and absolute inclusion of nutritional treatment, however, based on SGA only risk of malnutrition was detected. It is recommended to interpret/explain this statement why the SGA scale is better.

Authors: we agree with the reviewer regarding the fact that considering SGA  “a better method” is not accurate and could lead to misunderstandings, it is especially useful for prevention, but not for diagnosis, this point has been corrected in the revised version of our manuscript.

Reviewer 2 Report

Comments and Suggestions for Authors

The authors used different index to evaluate malnutrition i HF patients. They found different criteria had their own signficance and they compared with each other. I think the study is well performed. I only had one comments. The figure legend should better clarify the meaning of each box so that readers could better understand the essentials of the study. 

Author Response

We sincerely thank the Editor for the interest in our patient cohort and results. Following your suggestions, we re-evaluated our manuscript and focused on the comments of the Reviewers. We sincerely thank the Reviewers for their constructive comments, which we found very helpful towards improving the quality of our study. Accordingly, specific changes have been made in the manuscript, based on these comments, as it is described in detail below in a point-by-point description of the changes introduced, and on how Reviewer’s concerns were addressed. Changes within the manuscript are indicated in red.

Reviewer: The authors used different index to evaluate malnutrition i HF patients. They found different criteria had their own signficance and they compared with each other. I think the study is well performed. I only had one comments. The figure legend should better clarify the meaning of each box so that readers could better understand the essentials of the study. 

Authors: As suggested by the reviewer, figure legends have been improved in the revised version of our manuscript.

Reviewer 3 Report

Comments and Suggestions for Authors

General comments

The authors have some potentially interesting data. The paper is data heavy and this makes for a difficult read. The authors have multiple themes within the paper that are not always easy to disentangle. For example they have BIA data on all patients (I think) but DXA on only a small subsample. Clarity would be gained by removing the DXA data and concentrate on the BIA body composition data. The relative importance of GLIM versus SGA is not always made clear. Also statements such as at line 338 are confounded by not knowing whether the same patients approximate to the same groups by the different criteria. Some unsubstantiated statements are made , e.g line 437 "both methods are reliable" , I saw no reliability data.

The authors refer to BIVA but a BIVA analysis (plots) are never presented. simple BIA predictions are provided.

Line 86. How was this ample size determined? Was it simply opportunistic based on patient availability? If the latter please provide power calculation details for this sample size.

Line 92 DXA was performed in only 13 patients, was BIA performed in the remainder? If so were the body composition data combined? Table 4 shows a sample of 16.

Lines 103 to 108. Were these the output of the Akern algorithms? BIA is a predictive technique that requires careful standardization and the literature is replete with studies attempting to validate predictions. Are the Akern eqns valid for your population? If your DXA patients also had BIA how do the BC values compare?

Line 117. There is no such thing as "nutritional ultrasound". US can be used to provide nutritionally useful information, e.g. your AT assessments but this is not "nutritional US". Please use precise phrasing.

Table 1. Why is a P value provided for overall preferred food intake? there is a clear difference with respect to "soft food"

Tables 1 and 2. These are characteristics by GLIM  or SGA respectively for the whole sample. What is missing is the concordance between the two classifications. For example, do the 33 "well nourished" individuals fall into the "No malnutrition" group? If the groupings markedly differ then continuing to look at these groups separately (e.g., Table 3 is appropriate. But if they are virtually identical except for one or two individuals than this questions the need for the split analysis. Please provide the congruence between the two grouping as far as is possible recognizing that the classification schemes are not identical. If this is not possible (it should be) this point should be discussed anyway.

M/s throughout kg not Kg.

Table 3. It is incorrect to list BMI, FM, lean mass etc. Under "BIVA" These are body composition parameters that are estimated from the BIA data, BIVA is a graphical method of presenting R and Xc that provide an overview interpretation of body composition. Please state whether data are mean +/- SD or median and IQR.

Generally, TBW is larger than ECW and ICW. Why, since TBW=ICW +ECW? I appreciate that these are probably median values and therefore this is an inexact calculation but, e.g. well nourished TBW = 44.2 kg and ECW + ICW = 40.9 kg a difference of 3.3 kg or 7.4%. This seems large> Please check data.

Also FFM = lean + bone. The hydration of the FFM is provided by TBW/FFM. If we perform this calculation (e.g, for case above as above 44.2/63.4 =0.697), it would appear that all patients are dehydrated unusual in heart failure where edema is often present.  Please consider and comment. Note that excess fluid (edema) in heart failure will be interpreted by BIA algorithms as FFM. Since an assumed hydration is also implicit in DXA, edema will confound these data also but not necessarily to the same degree. See doi: 10.1080/17434440.2020.1791701

Section 3.4 OK so there were correlations between DXA and BIA but correlation is NOT agreement. What was the actual agreement in absolute values between the methods? This needs to be presented in order to assess comparison. Calculate absolute differences and determine LOA. This is important since it may shed light on the suitability of the BIA prediction equations in this population.

Line 368. The distinction between MFBIA and SFBIA devices is likely not to reflect the number of frequencies of measurement per se but the simply the different predictive algorithms used. The PREDICTIVE transformation method is the same unless the "MFBIA" device is actually a BIS device using mixture theory (e.g. doi: 10.3389/fcvm.2021.636718 or doi.org/10.1038/s41598-020-60358-y) The true distinction is not on frequencies but the transformation algorithms.

Author Response

We sincerely thank the Editor for the interest in our patient cohort and results. Following your suggestions, we re-evaluated our manuscript and focused on the comments of the Reviewers. We sincerely thank the Reviewers for their constructive comments, which we found very helpful towards improving the quality of our study. Accordingly, specific changes have been made in the manuscript, based on these comments, as it is described in detail below in a point-by-point description of the changes introduced, and on how Reviewer’s concerns were addressed. Changes within the manuscript are indicated in red.

REVIEWER 3

Reviewer:  General comments

The authors have some potentially interesting data. The paper is data heavy and this makes for a difficult read. The authors have multiple themes within the paper that are not always easy to disentangle. For example they have BIA data on all patients (I think) but DXA on only a small subsample. Clarity would be gained by removing the DXA data and concentrate on the BIA body composition data. The relative importance of GLIM versus SGA is not always made clear. Also statements such as at line 338 are confounded by not knowing whether the same patients approximate to the same groups by the different criteria. Some unsubstantiated statements are made , e.g line 437 "both methods are reliable" , I saw no reliability data.

Reviewer:  The authors refer to BIVA but a BIVA analysis (plots) are never presented. simple BIA predictions are provided.

Authors: As the reviewer mentioned, DXA is available in a subgroup of the patients, for avoiding confusion, this point was mentioned in more detail in the revised version of our manuscript. The importance of GLIM criteria and SGA has been revised in the discussion and the conclusion of the revised version of our manuscript, this point was also suggested by reviewer 1. Lines 338 and 437 have been rephrased.

Reviewer:  Line 86. How was this ample size determined? Was it simply opportunistic based on patient availability? If the latter please provide power calculation details for this sample size.

Authors: All patients that underwent a hospital admission due to HF were invited to participate, only those that accepted were included.

Reviewer:  Line 92 DXA was performed in only 13 patients, was BIA performed in the remainder? If so were the body composition data combined? Table 4 shows a sample of 16.

Authors: BIA was performed in all patients. DXA was performed in 16 patients, the typo in line 92 was corrected. We thank the reviewer for noticing this mistake.

Reviewer:  Lines 103 to 108. Were these the output of the Akern algorithms? BIA is a predictive technique that requires careful standardization and the literature is replete with studies attempting to validate predictions. Are the Akern eqns valid for your population? If your DXA patients also had BIA how do the BC values compare?

Authors: The Akern eqns are valid for our population, its characteristics have been detailed in the revised version of our manuscript.

Reviewer:  Line 117. There is no such thing as "nutritional ultrasound". US can be used to provide nutritionally useful information, e.g. your AT assessments but this is not "nutritional US". Please use precise phrasing.

Authors: Line 117 has been rephrased.

Reviewer:  Table 1. Why is a P value provided for overall preferred food intake? there is a clear difference with respect to "soft food"

Authors: An ANOVA was performed; a single p is provided.

Reviewer:  Tables 1 and 2. These are characteristics by GLIM or SGA respectively for the whole sample. What is missing is the concordance between the two classifications. For example, do the 33 "well nourished" individuals fall into the "No malnutrition" group? If the groupings markedly differ then continuing to look at these groups separately (e.g., Table 3 is appropriate. But if they are virtually identical except for one or two individuals than this questions the need for the split analysis. Please provide the congruence between the two grouping as far as is possible recognizing that the classification schemes are not identical. If this is not possible (it should be) this point should be discussed anyway.

Authors: the concordance was provided in the revised version of our manuscript, differences between methods have been also highlighted

Reviewer:  M/s throughout kg not Kg. 2

Authors: is has been corrected in the revise version of our manuscript.

Reviewer:  Table 3. It is incorrect to list BMI, FM, lean mass etc. Under "BIVA" These are body composition parameters that are estimated from the BIA data, BIVA is a graphical method of presenting R and Xc that provide an overview interpretation of body composition. Please state whether data are mean +/- SD or median and IQR.

Authors: Since only BIA parameters have been reported in this manuscript, the term BIVA has been removed in the revised version of our manuscript.

Reviewer:  Generally, TBW is larger than ECW and ICW. Why, since TBW=ICW +ECW? I appreciate that these are probably median values and therefore this is an inexact calculation but, e.g. well nourished TBW = 44.2 kg and ECW + ICW = 40.9 kg a difference of 3.3 kg or 7.4%. This seems large> Please check data.

Also FFM = lean + bone. The hydration of the FFM is provided by TBW/FFM. If we perform this calculation (e.g, for case above as above 44.2/63.4 =0.697), it would appear that all patients are dehydrated unusual in heart failure where edema is often present. Please consider and comment. Note that excess fluid (edema) in heart failure will be interpreted by BIA algorithms as FFM. Since an assumed hydration is also implicit in DXA, edema will confound these data also but not necessarily to the same degree. See doi: 10.1080/17434440.2020.1791701

Authors: Data is provided in median and IQR, for that reason, some differences in the sum are observed. Data has been checked. We thank the reviewer for pointing this out.

Reviewer:  Section 3.4 OK so there were correlations between DXA and BIA but correlation is NOT agreement. What was the actual agreement in absolute values between the methods? This needs to be presented in order to assess comparison. Calculate absolute differences and determine LOA. This is important since it may shed light on the suitability of the BIA prediction equations in this population.

Authors: As suggested by the reviewer, the absolute differences in fat mass and lean mass have been reported in the revised version of our manuscript.

Reviewer:  Line 368. The distinction between MFBIA and SFBIA devices is likely not to reflect the number of frequencies of measurement per se but the simply the different predictive algorithms used. The PREDICTIVE transformation method is the same unless the "MFBIA" device is actually a BIS device using mixture theory (e.g. doi: 10.3389/fcvm.2021.636718 or doi.org/10.1038/s41598-020-60358-y) The true distinction is not on frequencies but the transformation algorithms.

Authors: For avoiding confusion and as suggested by the reviewer, only body composition parameters determined by BIA were reported, thus, this has been corrected in the revised version of our manuscript.